# A conserved trypanosomatid differentiation regulator controls substrate attachment and morphological development in *Trypanosoma congolense*

**Eleanor Silvester[1]☉, Balazs Szoor[1]☉, Alasdair Ivens[1], Georgina Awuah-Mensah[2], Catarina Gadelha[2], Bill Wickstead[2], Keith R. Matthews[1]***

1 Institute for Immunology and Infection Research, School of Biological Sciences, University of Edinburgh, Ashworth laboratories, Charlotte Auerbach Road, Edinburgh, United Kingdom, 2 Medical School, Centre for Genetics and Genomics, School of Life Sciences, Queen's Medical Centre, University of Nottingham, Nottingham, United Kingdom

☉ These authors contributed equally to this work.
* keith.matthews@ed.ac.uk

**Data Availability Statement:** Gene expression data is available in the GEO database. The GEO

## Abstract

Trypanosomatid parasites undergo developmental regulation to adapt to the different environments encountered during their life cycle. In *Trypanosoma brucei*, a genome wide selectional screen previously identified a regulator of the protein family ESAG9, which is highly expressed in stumpy forms, a morphologically distinct bloodstream stage adapted for tsetse transmission. This regulator, TbREG9.1, has an orthologue in *Trypanosoma congolense*, despite the absence of a stumpy morphotype in that parasite species, which is an important cause of livestock trypanosomosis. RNAi mediated gene silencing of TcREG9.1 in *Trypanosoma congolense* caused a loss of attachment of the parasites to a surface substrate in vitro, a key feature of the biology of these parasites that is distinct from *T. brucei*. This detachment was phenocopied by treatment of the parasites with a phosphodiesterase inhibitor, which also promotes detachment in the insect trypanosomatid *Crithidia fasciculata*. RNAseq analysis revealed that TcREG9.1 silencing caused the upregulation of mRNAs for several classes of surface molecules, including transferrin receptor-like molecules, immunoreactive proteins in experimental bovine infections, and molecules related to those associated with stumpy development in *T. brucei*. Depletion of TcREG9.1 in vivo also generated an enhanced level of parasites in the blood circulation consistent with reduced parasite attachment to the microvasculature. The morphological progression to insect forms of the parasite was also perturbed. We propose a model whereby TcREG9.1 acts as a regulator of attachment and development, with detached parasites being adapted for transmission.

## Author summary

*Trypanosoma congolense* is a major cause of livestock trypanosomosis in sub-Saharan Africa where it contributes to lost economic productivity and poverty. These parasites

accession number for the RNA Seq sequence data is GSE249623.

**Funding:** This work is supported by a Wellcome Trust investigator award (Wellcome Trust, https://wellcome.org; grant number 221717/Z/20/Z to KM), a NC3Rs (https://nc3rs.org.uk) Research Grant (NC/W001144/1 to CG); a BBSRC (https://www.ukri.org/councils/bbsrc/) Research Grant (BB/W005867/1 to CG) and BBSRC (https://www.ukri.org/councils/bbsrc/) Research Grant (BB/W000342/1 to BW). The funders had no role in study design, data collection and analysis, decision to publish, or preparation of the manuscript.

**Competing interests:** The authors have declared that no competing interests exist.

differ in two key respects from the better-studied African trypanosomes, *Trypanosoma brucei*. Firstly, *T. congolense* adheres to surfaces in vitro and in the vasculature in vivo. Secondly, they lack a morphologically distinct transmission stage equivalent to the stumpy form of *T. brucei*. In the current work we identify a *T congolense* orthologue of a key developmental regulator in *T. brucei*, REG9.1, previously identified by a genome-wide RNAi screen. By RNA interference in transgenic *T. congolense* we demonstrate that TcREG9.1 functions both in parasite adherence in vitro and in vivo, and also in parasite development. The work provides a first insight into the unusual adherence and developmental biology of *T. congolense* and also suggests a possible model for how these characteristics interact to promote disease spread.

## Introduction

African trypanosome parasites cause substantial health and economic harms in sub–Saharan Africa through the diseases they cause in humans (*Trypanosoma brucei rhodesiense*, *Trypanosoma brucei gambiense*) and livestock (*Trypanosoma brucei brucei*, *Trypanosoma congolense*, *Trypanosoma vivax*) [1]. *T. brucei brucei* in particular has been the subject of intense molecular and cellular study and there is significant understanding of their mechanisms of immune evasion by antigenic variation [2,3], their cytological features [4,5] and their developmental biology [6–8]. Most African trypanosome species are spread by tsetse flies but *T. brucei* is unusual in showing extreme developmental adaptations in preparation for their transmission [9]. Specifically, this species exhibits two morphological types in the mammalian host—the slender and stumpy form—although there exists a spectrum of morphologies between these extremes such that *T. brucei* parasites are described as pleomorphic [10]. In contrast, the morphologies of *T. congolense* and *Trypanosoma vivax* parasites in their mammalian host are more uniform.

A great deal of work has focused on understanding the biology of *T. brucei* slender and stumpy forms and how the transition between these morphotypes is regulated by quorum sensing [11]. This has revealed that the development from slender to stumpy forms involves their regulation of a number of characteristic molecules, the commitment to cell cycle arrest of the parasites and thereafter morphological development to the stumpy cell form [12]. This development is important because it optimises the parasites for their successful colonisation of the tsetse midgut when *Trypanosoma brucei* parasites are ingested in a tsetse fly bloodmeal [13,14]. Among the key molecules that define the stumpy forms are PAD ('proteins associated with differentiation') proteins [15], and the expression of mRNAs comprising the ESAG9 family [16]. ESAG9 genes are sometimes but not always located in the expression sites that contain the variant surface glycoprotein genes used by the parasite for immune evasion [17,18]. The protein family is diverse in sequence, with some members predicted to possess a GPI anchor, and is of unknown function [19], but is predicted to encode surface or released proteins whose mRNAs are highly expressed as the parasites develop to stumpy forms [16]. As with many developmentally regulated genes in trypanosomes, analysis of the gene expression control of the ESAG9 family has revealed that posttranscriptional mechanisms are important [20,21]. This is because most *T. brucei* genes are encoded in multi-gene transcription units such that differential mRNA abundance is generated by distinct mRNA stabilities rather than transcriptional control mechanisms [22]. Exploiting this, a genome-wide gene silencing screen was previously performed to identify negative regulators of ESAG9 expression, and so possible regulators of the slender to stumpy transition [21]. This screen identified a RRM (RNA recognition motif) class predicted RNA binding protein designated TbREG9.1 (*T. brucei* Regulator

of ESAG9–1). Further characterisation of TbREG9.1 confirmed that the molecule was a repressor of ESAG9 transcripts, whose silencing increased the abundance of ESAG9 mRNAs. Moreover, silencing of TbREG9.1 resulted in an enhanced capacity for differentiation to stumpy forms in bloodstream form *T. brucei* and the altered expression of a number of developmentally regulated mRNAs in addition to those encoding ESAG9, this including several surface phylome family members (e.g. surface phylome Family 7 and Family 5 members; [19]). Interestingly, overexpression of TbREG9.1 induced bloodstream forms to undergo an inefficient but spontaneous differentiation to the next life cycle stage, procyclic forms. In combination these studies implicated TbREG9.1 as a regulator of gene expression and developmental events in *T. brucei* able to influence the expression of a stumpy form transcriptome profile and the capacity of the parasites for onward transmission.

In recent work, the mRNA expression of *T. congolense* parasites in the mammalian host has been characterised by RNAseq analysis [23]. This revealed that although *T. congolense* does not undergo morphological differentiation between slender and stumpy forms, the parasites undergo density-dependent cell cycle arrest and exhibit altered gene expression as parasite numbers accumulate in the host bloodstream, including the regulation of several predicted surface phylome families. Here we have explored whether this dynamic regulation of gene expression was controlled analogously to the regulation of stumpy mRNAs by TbREG9.1. This has identified a syntenic orthologue of TbREG9.1, TcREG9.1, whose silencing perturbs the expression of a number of molecules related to those characteristic of developmental adaptation in *T. brucei*. Most significantly, the silencing of TcREG9.1 was found to change the proportion of parasites that adhere to the surface of culture flasks and in vivo- a characteristic that is unique to *Trypanosoma congolense* and distinct from *T. brucei* that always remain in suspension in culture or free in the blood. Our results implicate TcREG9.1 as a regulator of adherence and development in *Trypanosoma congolense*, revealing that these parasites regulate their gene expression and adherence in the mammalian host potentially as an adaptation for transmission.

## Results

### Silencing TcREG9.1 reduces the substrate attachment of parasites

Analysis of the RNA expression profile of *T. congolense* in rodent infections revealed that several surface phylome families [19] (especially members of family 22) were elevated as parasites accumulated at the peak of the first wave of parasitaemia [23]. Our previous studies had demonstrated that a predicted RNA regulator, TbREG9.1 was able to control expression of *T. brucei* surface phylome families 5 and 7 as well as ESAG9 when parasites progressed to the stumpy form in vivo [21]. This led us to explore whether an equivalent regulatory process might be acting in *T. congolense* in response to increasing parasite density. Searching the *Trypanosoma congolense* EUPathDB genome sequence resource (https://tritrypdb.org/tritrypdb/app/) revealed the presence of a syntenic orthologue of TbREG9.1, TcIL3000.11.14540.1 (TcREG9.1) predicted to encode a protein of 519 amino acids, similar in size to TbREG9.1 (516 amino acids). The molecule encodes a predicted RRM domain (aa 195–244 in the *T. brucei* protein; e value $9.3 \times 10^{-5}$) that is more highly conserved between *T. brucei* and *T. congolense* than the remainder of the protein (Fig 1a; S1 Fig).

To test whether TcREG9.1 contributed to the regulation of *T. congolense* growth or development, the gene was silenced in *T. congolense* bloodstream forms by RNA interference. Specifically, the *T. congolense* IL3000 single marker line TcoSM, expressing the T7 RNA polymerase and the Tetracycline repressor protein, was generated by transfection with the pTcoSM plasmid, and cell lines selected with puromycin [24]. Thereafter, a region of TcREG9.1 was inserted into the *T. congolense* RNAi plasmid p3T7-TcoV [24] between

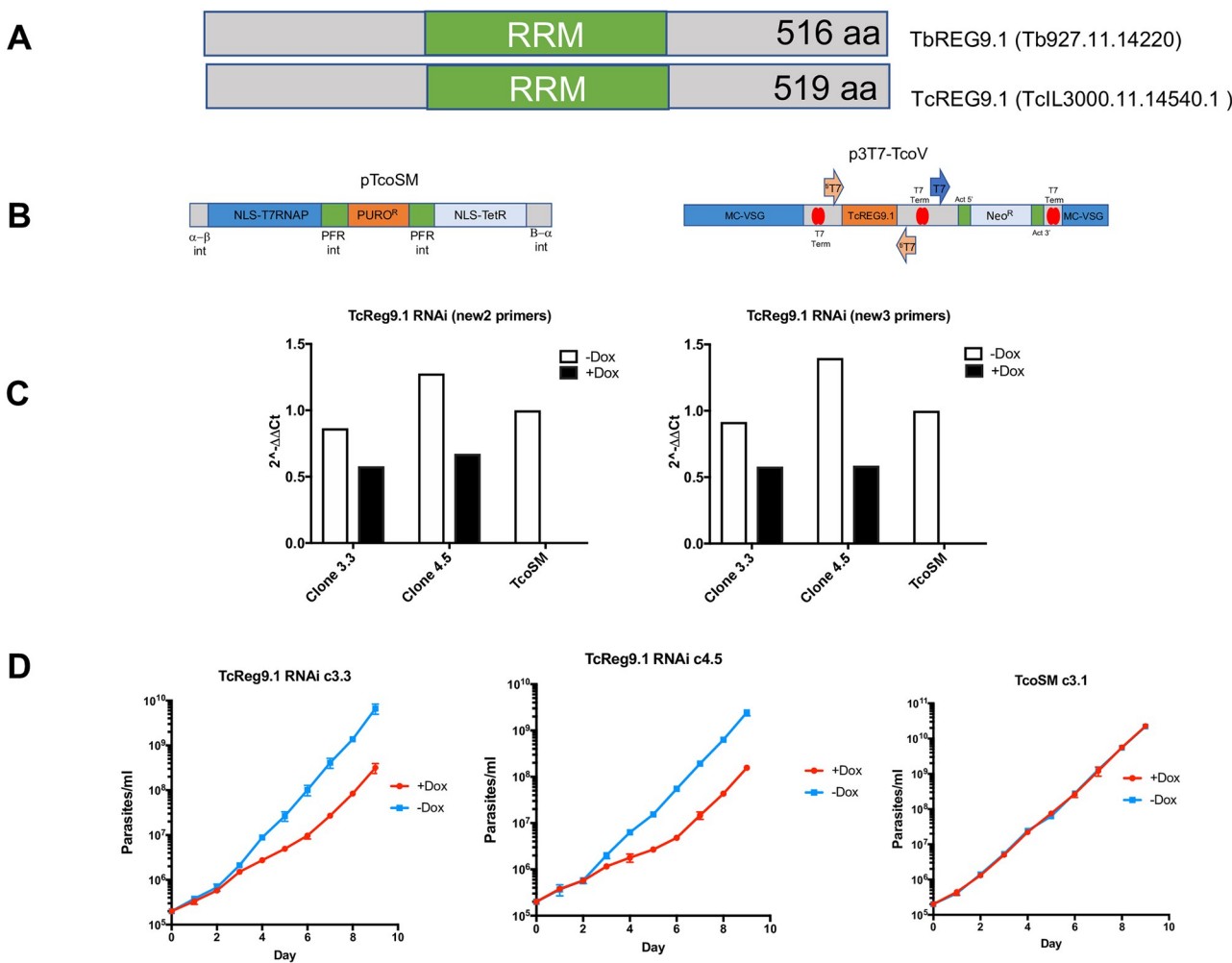

**Fig 1.** a. Schematic representation of TbREG9.1 and TcREG9.1 highlighting the position of the predicted RRM domain. b. Plasmid constructs used to engineer the RNAi depletion of the TcREG9.1 in *T. congolense* bloodstream forms. The TcoSM was developed according to [24], with TcREG9.1 sequence inserted in p3T7-TcoV. c. qRT-PCR of the mRNA levels of TcREG9.1 in two derived RNAi lines, clone 3.3 and clone 4.5, plus the parental line TcoSM. With induction using doxycycline TcREG9.1 levels were depleted approximately 50% in comparison to uninduced or parental cells. d. Growth in vitro of *T. congolense* lines (TcREG9.1cl3.3; TcREG9.1 cl4.5) in the presence (red) or absence (blue) of doxycycline in order to deplete TcREG9.1 levels in the RNAi lines. TcoSM parental cells were also grown ±doxycycline.

opposing T7 RNA polymerase promoters (Fig 1b) and transfected into the TcoSM cell line under G418 selection, resulting in the isolation of cell lines capable of TcREG9.1 gene silencing under doxycycline or tetracycline induction. A number of resulting clones were isolated after selection and for two of these (clone 3.3, clone 4.5) the level of gene silencing was analysed by qRT-PCR using distinct primer pairs, confirming inducible knock down of the target TcREG9.1 mRNA albeit with approximately ~50% of the level of mRNA in uninduced cells remaining (Fig 1c).

Analysis of clones 3.3 and 4.5 demonstrated that each showed a reduction of their growth upon TcREG9.1 depletion this being detected 3 days after the induction of RNAi with doxycycline (Fig 1d). Interestingly, in addition to a reduced growth rate a further phenotype was evident. In culture, *T. congolense* (unlike *T. brucei*) is characterised by attaching to the bottom of the plastic culture flask such that parasites are visible both attached and in the culture

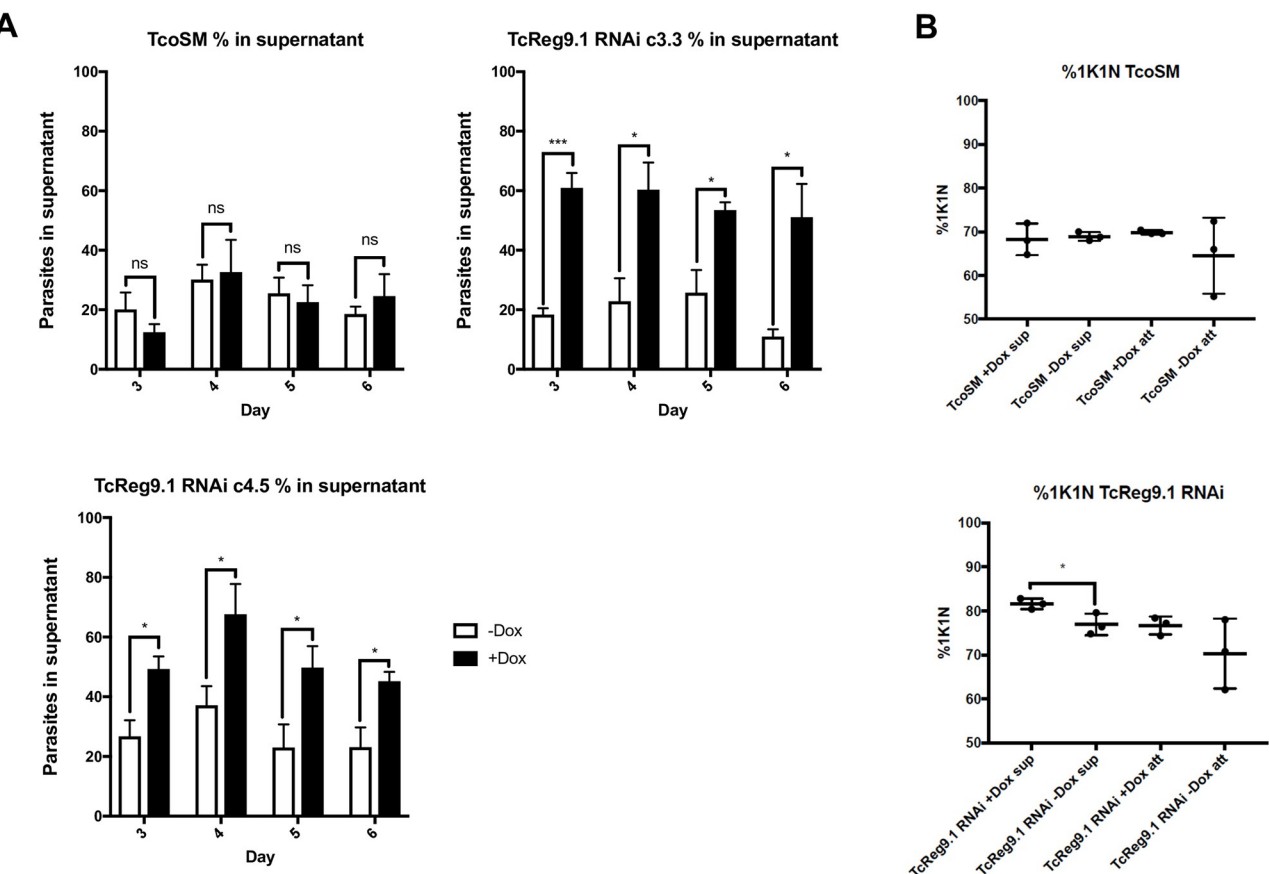

**Fig 2.** a. Proportion of *T. congolense* parasites that were free swimming in the culture supernatant in control TcoSM cells or at time points after the induction of TcREG9.1 depletion by RNAi in clones TcREG9.1cl3.3 or TcREG9.1cl4.5. b. The percentage of parasites exhibiting a 1 kinetoplast 1 nucleus (1K1N) organelle configuration (representing cells in G0, G1 and S phase) whether attached ('att') or detached ("sup") from the culture flask for control TcoSM cells (upper panel) or with doxycycline induced depletion of TcREG9.1 (lower panel). Samples were prepared 3 days after addition of doxycycline. In both attached and detached cells, depletion of TcREG9.1 resulted in a higher proportion of 1K1N cells, this being significant for the detached population (*; P<0.05).

supernatant, this reflecting their reported adherence to the vasculature in vivo [25]. Upon depletion of TcREG9.1 by RNAi the proportion of attached cells decreased and more parasites were present in the supernatant in detached form (Fig 2a). This was not evident in the control TcoSM parental lines upon doxycycline exposure, preceded the growth phenotype, and resulted in two-three times as many parasites in the supernatant in the induced populations compared to those that were uninduced, regardless of the clone analysed. Further analysis of the cell cycle profile of the induced and uninduced parasites indicated a small but significant (p<0.05) increase in the proportion of detached parasites exhibiting a 1 kinetoplast 1 nucleus configuration of DNA containing organelles, indicative of accumulation in the G0/G1/S phase of the cell cycle (Fig 2b). Hence, upon TcReg9.1 gene silencing, parasites demonstrate a reduced adherence phenotype, slowed growth and with an increase in the proportion of cells with 1K1N in the supernatant population.

## The detachment phenotype is phenocopied by PDE inhibitors

The detachment phenotype observed upon the silencing of TcREG9.1 was reminiscent of a similar phenomenon observed in the insect trypanosomatid *Crithidia fasciculata* treated with

phosphodiesterase inhibitors [26]. Specifically, like *T. congolense*, *Crithidia* is characterised in culture by attachment to the culture flask and recent studies have demonstrated that increasing cellular cAMP by inhibiting the action of phosphodiesterase using the inhibitor compound A (NPD-001) [27] generated increased levels of detached parasites. To explore whether treating *T. congolense* parasites with compound A also resulted in detachment, we quantitated the level of attached and detached TcoSM parasites after cells were incubated for 60 minutes in culture wells and then exposed to different concentrations of compound A. Fig 3a demonstrates that parasites treated with 10µM compound A showed fewer attached cells after 24h compared to control cells. Interestingly, compound A exposure was accompanied by more vigorous motility of the treated cells, generating enhanced migration of the parasites in the presence of 10µM of the drug after 2h (Fig 3b and 3c), enabled by their detachment. To explore the basis of this, parasites were exposed to 10µM compound A and their motility tracked at different time

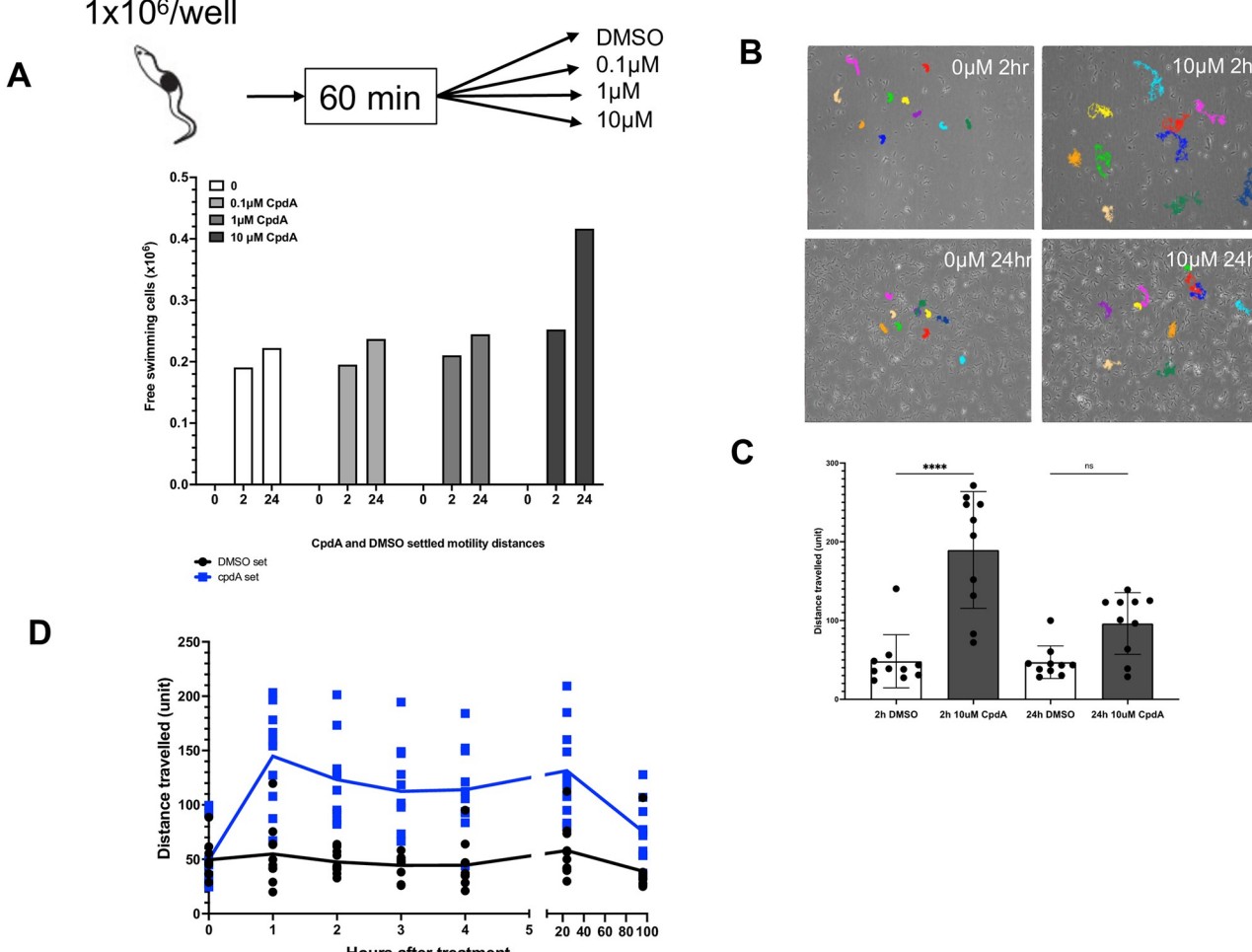

**Fig 3.** a. Schematic representation of the experimental regimen. Free swimming parasites were aliquoted in to culture wells, left for 60 minutes to attach and then exposed to different concentrations of compound A, or DMSO. Below is shown the proportion of attached and detached parasites at 0h, 2h and 24h in 0, 0.1µM, 1µM and 10µM compound A. b. Motion tracks of parasites exposed to DMSO, or 10µM compound A for 2h or 24h. In each case individual cells (represented by different colour tracks) are shown, with compound A exposure resulting in greater migration reflecting increased vibrancy in their motility. c. Quantitation of the motion tracks of parasites exposed to DMSO, or 10µM compound A for 2h or 24h. 10 cells in each condition were tracked. There was enhanced vibrancy in parasites exposed to 10µM compound A for 2h (***; P<0.01). d. Time course of cell vibrancy after exposure to CpdA (blue) or DMSO (black).

points. This revealed that the parasites exhibited enhanced motility with PDE inhibition by compound A for 24-48h (Fig 3d). In combination, these assays demonstrated that *T. congolense* attachment and detachment can be regulated by PDE inhibition within the parasites pharmacologically, similar to *Crithidia fasciculata*, and that REG9.1 depletion may operate through a shared pathway to this.

## TcREG9.1 alters the abundance of transcripts associated with development in the mammalian host

Given its anticipated effect on posttranscriptional gene regulation through its RRM RNA binding module, we examined the consequences of TcREG9.1 depletion for the gene expression profile of *T. congolense* parasites. Thus, three replicates of each of the TcREG9.1 RNAi lines cl3.3 and cl4.5 were independently induced to silence TcREG9.1 by growth with doxycycline for 3 days, with uninduced replicates grown in parallel (Fig 4a). This timepoint was chosen as it was prior to the growth reduction that accompanies this gene silencing, thereby allowing transcript changes associated with TcREG9.1 depletion rather than only slow growth to be identified. As before, detachment of the parasites was detected in this timeframe of induction, such that for both clones, approximately twice as many parasites were detected in the culture supernatant when TcReg9.1 was depleted (Fig 4b). RNAs were isolated from the attached and detached parasites combined for each of the replicate cultures, both induced and uninduced,

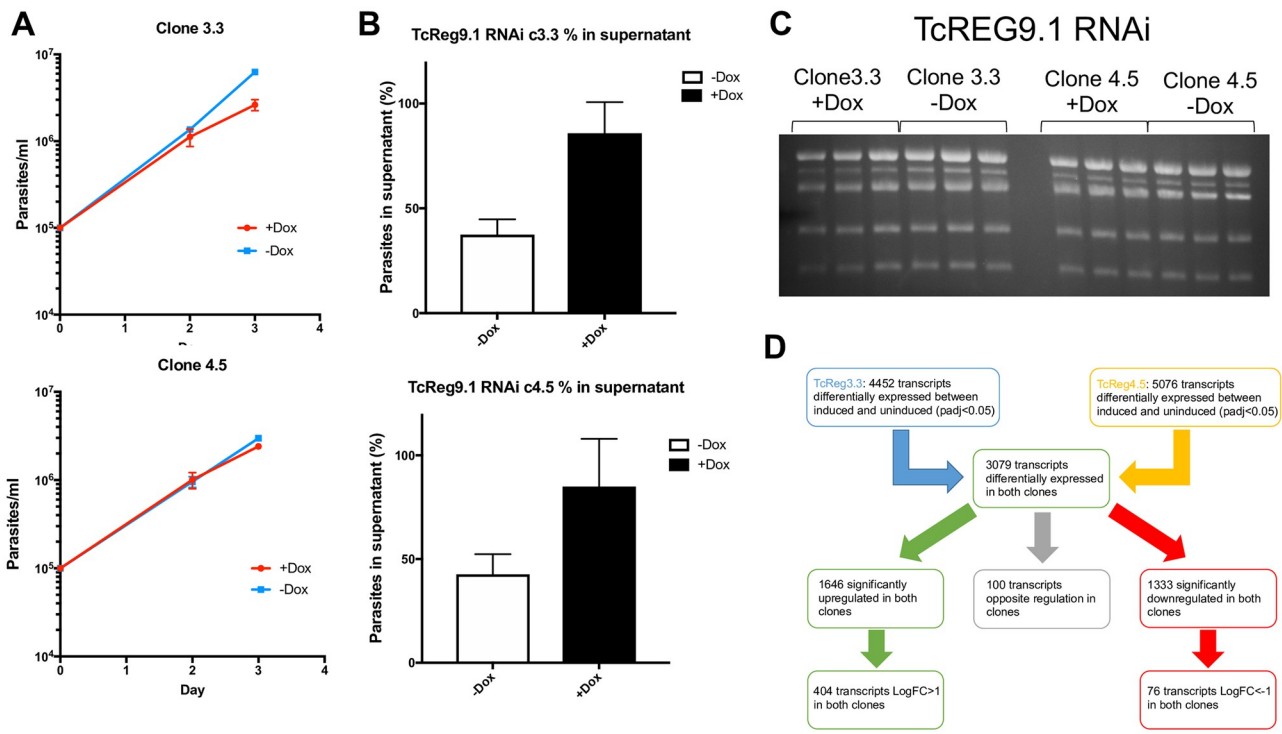

**Fig 4.** a. Growth of TcREGcl3.3 and TcREGcl4.5 in the presence and absence of doxycycline over 72h. Triplicate growth profiles are shown with error bars representing ±SEM. b. Samples subject to TcReg9.1 RNAi were analysed for their adherence to the culture flask to validate increased levels of parasites in the supernatant after TcREG9.1 depletion. Three biological replicates are shown with error bars representing ±SEM. c. Isolated RNA derived from the cultures represented in panel a and b, with the rRNA bands (*T. congolense* has 5 major rRNA bands) visualised on an ethidium bromide stained gel. For each condition, biological triplicates were analysed and RNA generated. d. Analysis pipeline for the transcripts derived with TcREG9.1. depletion induced or not after doxycycline treatment with the number of transcripts identified as being upregulated or downregulated being annotated.

generating a total of twelve samples (Fig 4c), which were subject to RNAseq analysis. Overall, 4452 transcripts were differentially regulated between induced and uninduced samples of TcREG9.1 RNAi clone 3.3 and 5076 transcripts were differentially regulated between induced and uninduced samples of TcREG9.1 RNAi clone 4.5 (Fig 4d; S1 Data). Of these differentially expressed RNAs, 3079 transcripts were shared between both clones, and their respective expression in each clone plotted on Fig 5. This demonstrated excellent consistency between the transcripts regulated upon TcREG9.1 RNAi in each clone, with 1646 transcripts being

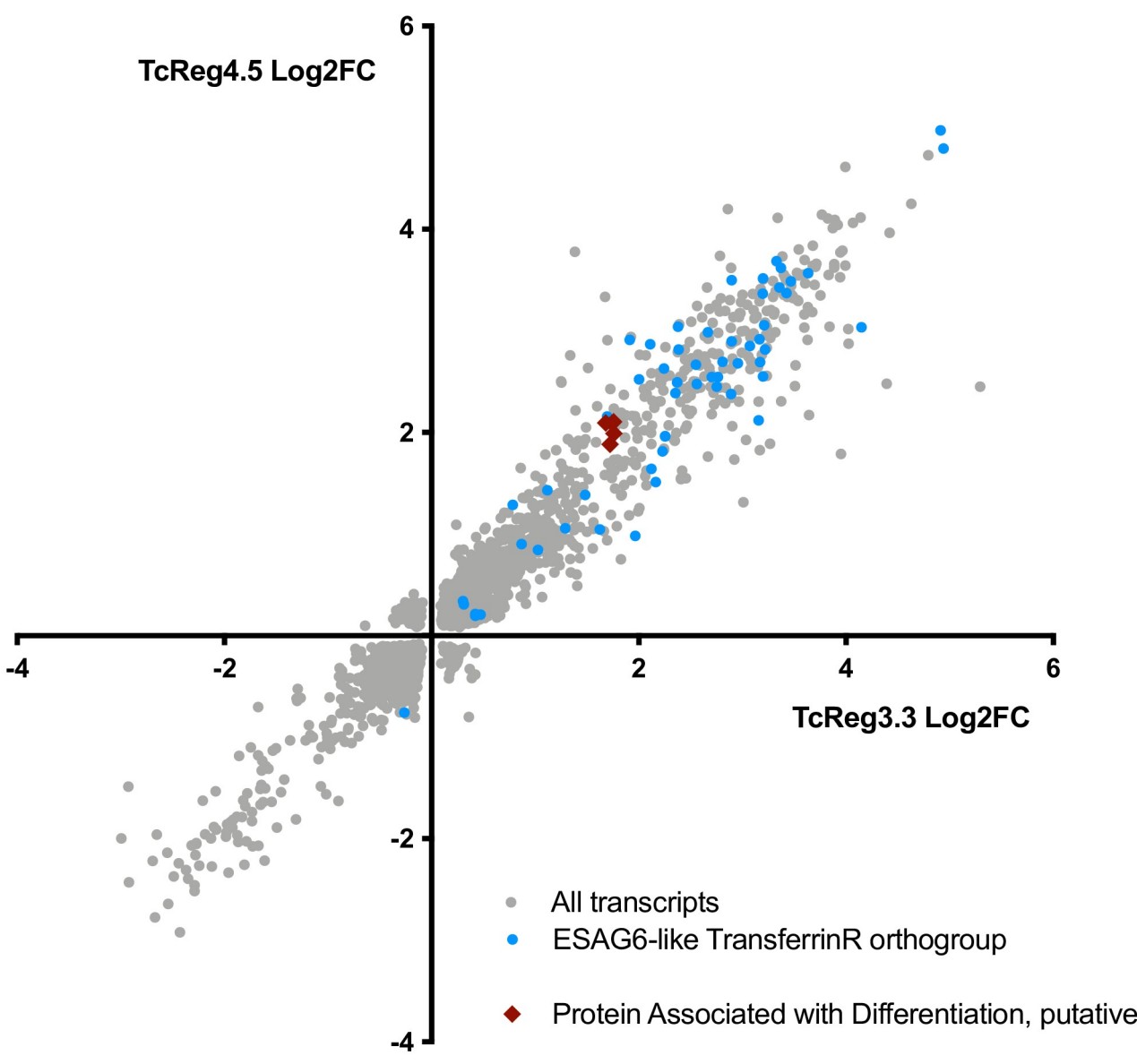

**Fig 5. Scatterplot of transcripts (p adj<0.05) that are regulated in the TcREG9.1 cl3.3 and TcREG9.1cl4.5 cell lines.** Transcripts with an annotation of Transferrin receptor/ESAG6 are highlighted in blue whereas those related to *T. brucei* PAD proteins ('*Proteins associated with differentiation*') are highlighted in red. Both groups of molecules are upregulated upon TcREG9.1 depletion.

significantly upregulated in both clones, 1333 being significantly downregulated in both clones and only 100 transcripts showing a change in abundance in the opposite direction between clones. Of the consistently regulated transcripts, 404 upregulated transcripts exhibited a LogFC>1 and 76 downregulated transcripts exhibited a LogFC<-1, consistent with our previous demonstration that the *T. brucei* orthologue of TcREG9.1 is a repressor of gene expression [21]. This focussed our attention on those mRNAs that become upregulated (adj P<0.05) upon the depletion of TcREG9.1. In this group were many transcripts encoding molecules previously observed to be elevated as the parasitaemia increased in vivo [23]. For example, at high parasitaemia in rodent infections we have previously observed significant upregulation of transcripts derived from genes that were related to the ESAG6/transferrin receptor family (Surface phylome family 15) (Fig 5, blue symbols; Fig 6, with transcripts in each surface phylome family colour-coded for their adj p<0.05 in each clone) [23]. Also strongly upregulated were some members of the Surface phylome family 22, which were previously observed to be strongly upregulated in peak parasitaemia parasites isolated from in vivo infections [23]. Genes annotated as related to those encoding the PAD family of proteins in *T. brucei* were also elevated upon TcREG9.1 depletion (Fig 5, red symbols), consistent with the elevated abundance of transcripts characteristic of parasites equivalent to stumpy forms in *T. brucei*.

As well an increased abundance of transcripts that are elevated at peak parasitaemia in vivo, TcREG9.1 depletion also caused an increased abundance of transcripts reported to encode immunoreactive antigens. In a previous study, the sera from bovine infections was used to select immunoreactive proteins expressed by *T. congolense* [28]. From this analysis proteins of several families were detected, notably members of the PLAC8 family, cathepsin L like cysteine peptidases, in addition to the ESAG6/transferrin receptor/PAG2 protein family and VSG and ISG family proteins (S2 Fig, S1 Table). Of these, there was evidence for a strong elevation of transcripts following TcREG9.1 depletion for several PLAC8 domain transcripts (Surface phylome family 75), some members of the cathepsin-L cysteine peptidase family (Surface phylome family 67), and ISGs (Surface phylome family 49) as well as the ESAG6 family proteins (Fig 6). Regulation in the transcriptome profiles was also observed for amino acid transporter (family 54) and purine nucleoside transporter family (family 61) genes, although there was evidence of both significant (adj p<0.05) upregulation and downregulation of distinct family members (Fig 6).

## TcREG9.1 depletion increases the proportion of circulating parasites in vivo

Unlike *T. brucei*, *T. congolense* adheres to the walls of the blood vasculature [25]. To investigate whether the depletion of TcREG9.1 reduced the level of adherent parasites in vivo, matching the reduction in attached parasites in culture flasks in vitro, mice were infected with *T. congolense* RNAi line cl3.3. Once parasites were detected by tail snips of 8 infected animals, doxycycline was provided in the drinking water to half of the infected group and then the parasitaemia allowed to progress for a further 3–4 days. Thereafter, the ratio of parasites detectable by tail snip analysis was compared to the number of parasites detected in the circulatory bloodstream after cardiac puncture. This provides a measure of the relative attachment/detachment of the parasites, since the parasites detected in the tail snips are enriched for parasites trapped in the small vessels of the tail [25] whereas circulating parasites are detached. Fig 7a–7c demonstrates that with TcREG9.1 depletion, there was an increased proportion of parasites detected in the cardiac blood samples compared to uninduced samples (p<0.001). Visually the parasites detected either in tail snip or cardiac samples and between induced and uninduced samples were indistinguishable. Thus, as well as detachment in vitro, depletion of

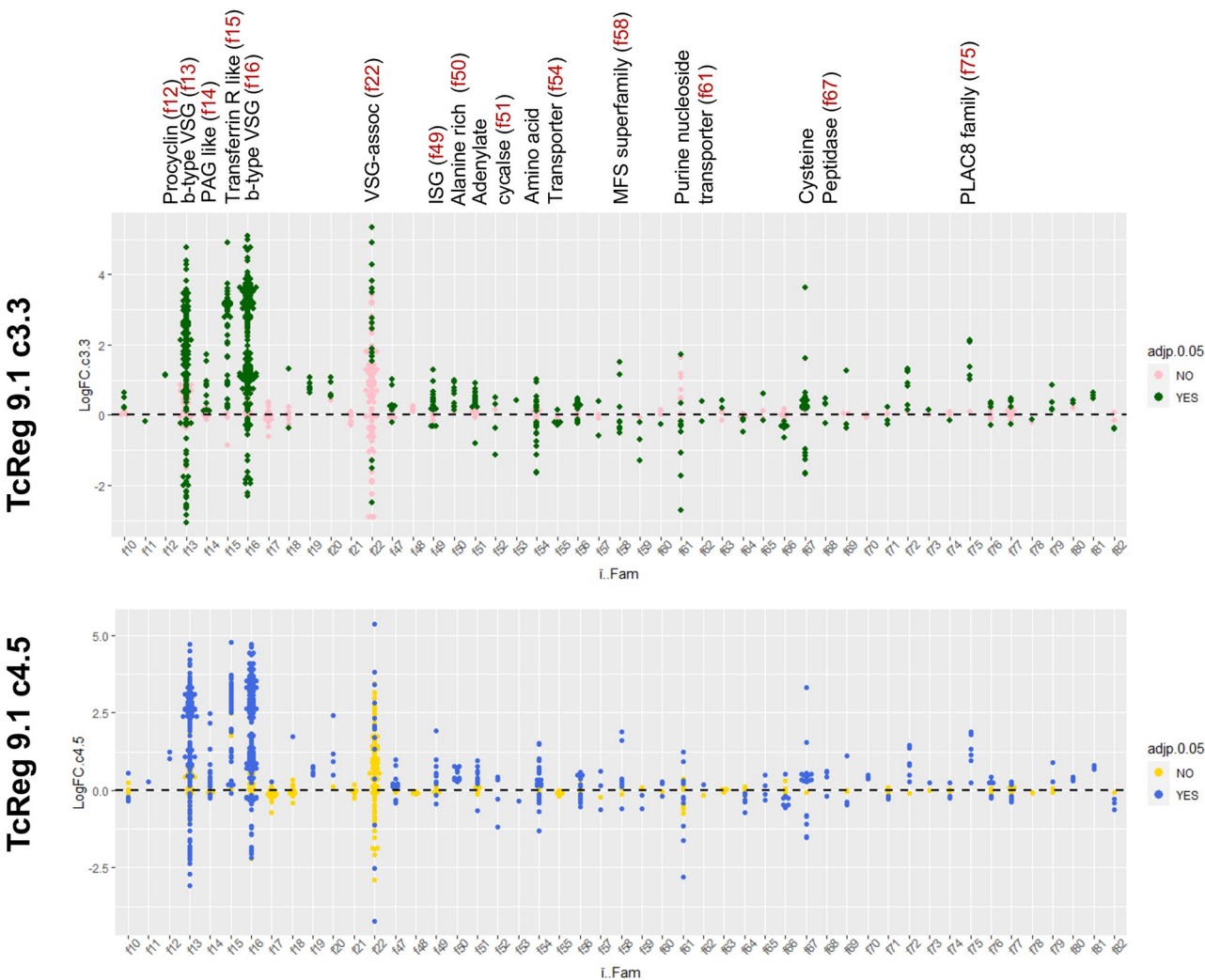

**Fig 6. LogFC enrichment of transcripts regulated upon TcREG9.1 depletion in the TcREG9.1cl3.3. and TcREG9.1cl4.5 cell lines, grouped by their surface phylome category (f10 to f82).** Individual transcripts are colour coded depending upon the statistical significance (Adj p<0.05) in each cell line; upregulated transcripts have a positive LogFC value, downregulated transcripts have a negative LogFC value. The most prominently regulated transcript groups are annotated along with their surface phylome family number above the upper plot.

TcREG9.1 results in a higher proportion of circulatory parasites in infected animals, indicative of reduced attachment in vivo.

## TcREG9.1 facilitates developmental cis aconitate independent kinetoplast repositioning

To explore whether there was a difference in the relative developmental capacity of *T. congolense* after TcREG9.1 depletion, we investigated their differentiation to tsetse midgut forms. To monitor differentiation, we harvested parasites and exposed them to the differentiation trigger cis aconitate and scored the distance between the kinetoplast and the posterior end of the cell (Fig 8a–8c). This distance increases as parasites differentiate to tsetse midgut forms in both *T. brucei* [29] and *T. congolense* [30]. Parasites were also monitored with an antibody to PRS, which labels insect stage parasites but not bloodstream forms [31]. Fig 8b demonstrates that

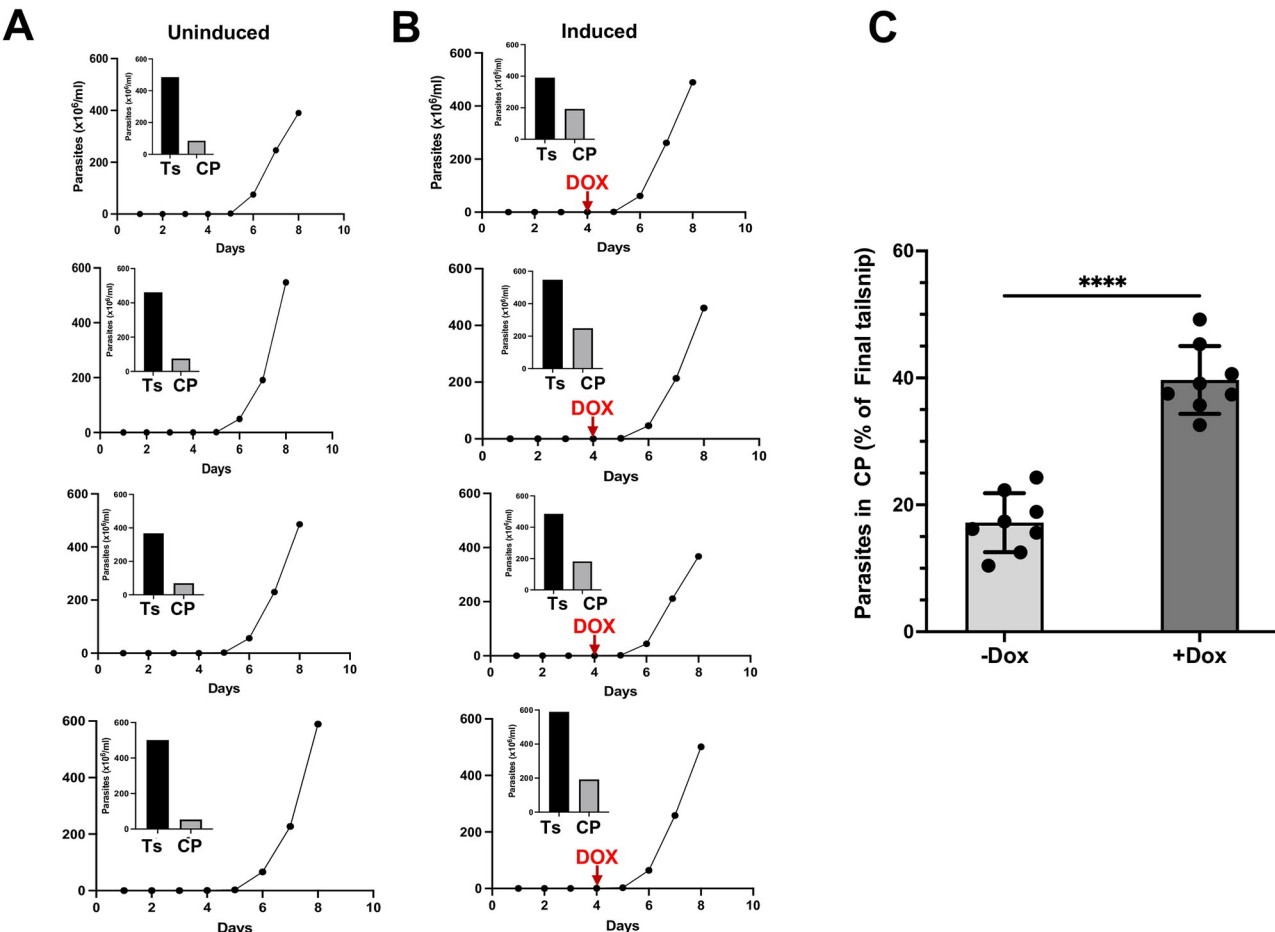

**Fig 7.** a. Parasitaemia of mice infected with the TcREG9.1 RNAi cell line, with the parasites detected on day 8 in the tail snip or blood from cardiac puncture represented in the inset bar chart. TcREG9.1 RNAi was uninduced. Each panel represents the parasitaemias from a distinct mouse. b. Parasitaemia of mice infected with the TcREG9.1 cell line, with the parasites detected on day 8 in the tail snip or blood from cardiac puncture represented in the inset bar chart. TcREG9.1 RNAi was induced by the addition of Doxycycline to the drinking water of mice on day 4 post infection. Each panel represents the parasitaemias from a distinct mouse. c. Proportion of parasites detected in the blood derived from a cardiac puncture of mice infected with parasites either induced or not to deplete TcREG9.1 by RNAi. Values are expressed relative to the proportion of parasites detected in tail snips. The data represents 8 individual infections each for parasites induced or not to deplete TcREG9.1.

the mouse-derived parasites responded to cis aconitate after 24h by increasing their kinetoplast-posterior dimension and expressing PRS, but the response was much less in culture-derived parasites. At this time point, no effect of TcREG9.1 depletion was seen. After 48h (Fig 8c) in DTM medium at 27˚C in the presence of cis aconitate, further kinetoplast repositioning had occurred with PRS expression, and this was regardless of TcREG9.1 depletion by RNAi. Interestingly, in the absence of cis aconitate, kinetoplast repositioning had also occurred at this timepoint albeit without the concomitant expression of PRS, demonstrating these events are separable. However, this PRS-independent kinetoplast repositioning was dependent upon TcREG9.1, being significantly reduced in parasites where TcREG9.1 was depleted by RNAi induction ($p < 0.0001$; statistical comparisons for all samples are provided in S2 Table). This effect was consistent for parasites derived from each independent mouse infection (4 with RNAi induced and 4 without RNAi induction) (S3 Fig). Hence, kinetoplast repositioning at 27˚C in DTM is cis aconitate independent and reduced in the absence of TcREG9.1. In

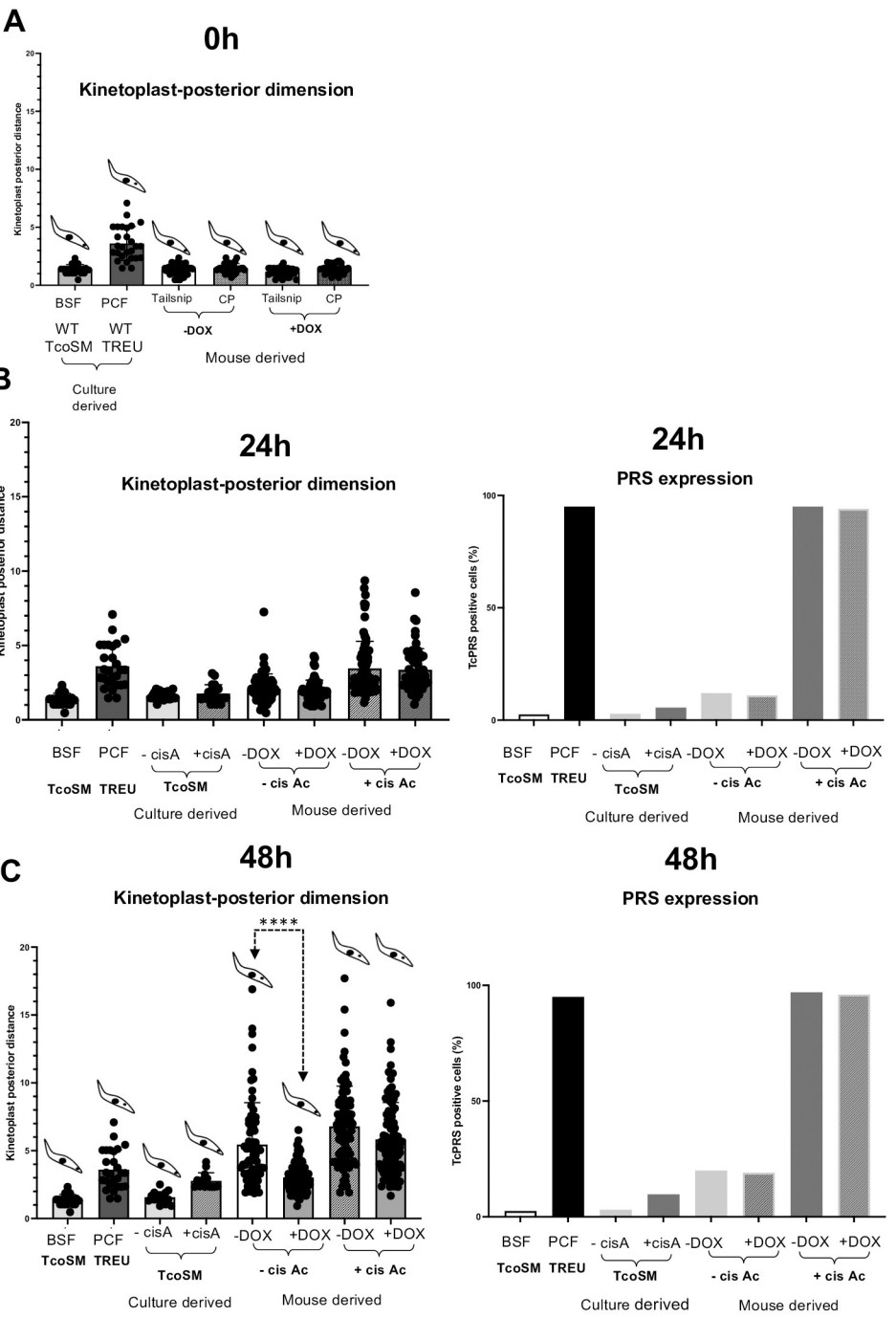

**Fig 8.** a. Kinetoplast-posterior dimension (a.u.) for bloodstream or procyclic form *T. congolense* in culture, and the same dimension for parasites derived from in vivo infections via tail snip or cardiac puncture with or without RNAi induced silencing of TcREG9.1. A schematic representation of the parasites is shown with the position of the cell nucleus and kinetoplast shown with respect to the cell posterior. b. Kinetoplast-posterior dimension (a.u.) for bloodstream or procyclic form *T. congolense*, or for parasites incubated for 24h in DTM medium at 27˚C with or without the differentiation trigger cis aconitate. Mouse derived parasites had been isolated from infections where TcREG9.1 RNAi was either induced or not. The proportion of parasites expressing PRS is shown at the right hand side. c. Kinetoplast-posterior dimension (a.u) for bloodstream or procyclic form *T. congolense*, or for parasites incubated for 48h in DTM medium at 27˚C with or without the differentiation trigger cis aconitate. Mouse derived parasites had been isolated from infections where TcREG9.1 RNAi was either induced or not. A schematic representation of the parasites is shown with the position of the cell nucleus and kinetoplast shown with respect to the cell posterior. The effect of RNAi against TcREG9.1 is highlighted by the arrows, indicating reduction in the kinetoplast -posterior

dimension with doxycycline but in the absence of cis aconitate (****; p<0.0001). The proportion of parasites expressing PRS is shown at the right hand side.

contrast, cis aconitate overrides the requirement for TcREG9.1 and permits both kinetoplast repositioning and PRS expression.

## Discussion

A key distinction between *Trypanosoma brucei* and *Trypanosoma congolense* is that *T. brucei* remains free swimming in its mammalian host, whereas *T. congolense* exhibits attachment to the epithelium of blood vessels. In particular *T. congolense* parasites are concentrated in the microvasculature, generating 4–1400 fold higher levels of parasites when assayed via tail snip compared to those detected in cardiac blood [25]. This phenomenon is also reflected by parasites in vitro where *T. congolense* proliferate attached to the culture flask, albeit with free swimming trypanosomes also prevalent. In the work described here we have perturbed the expression of an orthologue of a key regulator, REG9.1, previously identified in *T. brucei* as important in the control of transmission stage development, and observed an inducible loss of adherence in vitro. Moreover, analysis of infections in vivo provided evidence that the same phenomenon occurs, generating a higher proportion of circulatory parasites upon the depletion of TcREG9.1. This was also accompanied by the altered expression of several predicted surface phylome families, these being reproducibly regulated in independent cell lines capable of inducible REG9.1 silencing. Additionally, a number of transcripts and transcript families diagnostic for the developmental progression of *T. brucei* and *T. congolense* were elevated upon the depletion of TcREG9.1, and TcREG9.1 depletion reduced developmental kinetoplast repositioning unless overridden by the presence of cis aconitate. Overall, this supports the molecule acting in *T. congolense* analogously to *T. brucei* by repressing the generation of transmission adapted parasites and offers first insight in to the molecular regulation of *T. congolense* adherence and developmental progression.

A number of previous studies have explored the attachment of kinetoplastid parasites to their substrate. With respect to the insect stages of the life cycle, the attachment of *T. congolense* has been characterised as being mediated via a flagellum-based attachment plaque which was resistant to non-ionic detergents and so not membrane dependent. Although the molecular nature of the attachment plaque was not defined, a prominent 70kDa protein was present in the plaque-enriched fractions [32]. Attachment has also been characterised in the fish trypanosomatid *Trypanosoma carrissii*, visualised in zebrafish as being mediated via the posterior end of the cell, with the flagellum remaining free and able to beat, distinguishing the attachment mode from that of *T. congolense* [33]. More detail has been determined for the attachment of the insect trypanosomatid, *Crithidia fasciculata*. Here, the parasites are, as with *T. congolense*, attached via the flagellum as haplomonads which exhibit a somewhat faster rate of proliferation than the detached nectomonad forms. This is similar to our observations of *T. congolense* in culture, where following TcReg9.1 depletion, the free-swimming parasites are enriched in parasites with 1 kinetoplast and 1 nucleus, whereas proliferation is most evident with the attached forms. In recent work, the attachment and detachment of the *Crithidia fasciculata* parasites could be altered by exposure to the phosphodiesterase inhibitor compound A (NPD001). This acts to perturb cyclic AMP levels in the cells by inhibiting phosphodiesterase action, and has been previously employed to explore the importance of cyclic nucleotide signalling in *T. brucei* [34]. As with *Crithidia*, we observed that *T. congolense* showed reduced adherence in response to compound A. This suggests that the attachment phenomenon in

both species may be regulated equivalently, although whether this is direct or indirect is unclear at present. This is because we observed that the *T. congolense* parasites showed greater vibrancy in their swimming in response to compound A treatment, opening the possibility that their lack of adherence was a consequence of the parasites either failing to stably attach or becoming forcibly detached by the beat vigour of the flagellum. Nonetheless, the flagellum and cAMP-based sensing is a key feature of *T. brucei* social motility that is important to the migration of that species through the tsetse fly [6,35]. Hence, involvement of this signalling control of the attachment organelle in *T. congolense* would not be unexpected.

In *T. brucei*, TbREG9.1 was identified as a repressor of gene expression that when silenced allowed the upregulation of ESAG9 transcripts that are characteristic of transmissible stumpy forms in that species [16,21]. This is anticipated to act via posttranscriptional control mechanisms given the RRM domain that is present in the molecule and also well conserved in the *T. congolense* orthologue. Moreover, an analysis of the regulatory activity of the molecule in protein tethering assays designed to report on negative or positive effects on gene expression revealed that in *T. brucei*, TbREG9.1 was able to repress gene expression [36]. A similar mechanism is likely to operate in *T. congolense* and we observed that there was significant upregulation of several transcript classes when TcREG 9.1 was silenced. Of note, this resulted in the significantly altered expression of a number of predicted surface phylome families, including the ESAG6/PAG2 surface phylome members, invariant surface glycoproteins (ISGs) and the VSG gene associated family 22 surface phylome members. Although ESAG6 is VSG expression site associated in *T. brucei*, in *T. congolense* the transferrin family of molecules is encoded outside of expression sites [19] and broadly expressed, being significantly upregulated at the peak of parasitaemia [23]. Many but not all of the encoded molecules of this family are also predicted to be GPI anchored [23], consistent with *T. brucei* where ESAG6 and 7 form a heterodimeric receptor of which only one is GPI anchored; many but not all ESAG9 proteins are also predicted to be GPI anchored and are regulated by TbREG9.1 [16]. Other predicted surface protein transcripts regulated by TcREG9.1 include ISGs, some of which have been found to be important in protecting bloodstream *T. brucei* against innate immune mechanisms [37], whereas the family 22 group of molecules are unusual genes which are predicted to be located in or close to the 3'UTR of VSG genes [19] and may or may not be translated into protein. In addition to these major families, we observed a consistent upregulation of PLAC8 family proteins. These are of unknown function but detected as immunoreactive antigens when the serum from *T. congolense* infected cattle were assayed for their IgG specificity [28], confirming their expression as proteins. In combination, this analysis indicates that many classes of predicted surface proteins are upregulated by the silencing of TcREG9.1 consistent with its action in *T. brucei*. Predicted major surface molecules have also been observed as transcriptome differences between adherent and free swimming *Crithidia fascicula*ta, with different gp63 members elevated in each form [38]. The diversity of the respective gene families regulated by TcREG9.1 limits the analysis of any contribution each may make to the adherence phenotype by gene silencing. Indeed, they may act individually or in combination by actively reversing adherence, or passively blocking successful adherence by occluding the binding of uncharacterised adherence mediators.

The regulation of the transferrin receptor/ESAG6/7 proteins in *T. congolense* upon TcREG9.1 silencing matched the upregulation for these transcripts when *T. congolense* progresses from ascending to peak parasitaemia's infection in vivo [23]. This development shows some similarities to the quorum-sensing based differentiation of *T. brucei* from proliferative slender forms to non-proliferative transmissible stumpy forms, involving the parasites accumulating in G1/GO, although morphological transformation is not a feature of *T. congolense* development [39]. Similarly, with the silencing of TcREG9.1 and detachment of the parasites,

there was upregulation of molecules related to the PAD protein family, which in *T. brucei* is diagnostic for the development to stumpy forms. Importantly, the loss of adherence was not only observed in culture flasks but also seen in vivo, with there being an enhanced level of circulating parasites isolated in cardiac bleeds upon TcREG9.1 silencing compared to where the silencing was not induced. This suggests that TcREG9.1 might share a similar role to TbREG9.1 in regulating the generation of *T. congolense* transmission stages, whose development would be accompanied by the detachment of the parasites from the endothelium of the vasculature. In support of this, depleting TcREG9.1 reduced one element of the development to insect forms, the repositioning of the kinetoplast, although TcREG9.1 dependency was overridden by the differentiation trigger cis aconitate.

Bringing together the observations after perturbation of TcREG9.1 expression generates an intriguing model for the proliferation and development of *T. congolense* (Fig 9). This would entail the multiplication of *T. congolense* parasites in the mammalian host while attached in the peripheral microvasculature, with parasites released in to the circulation upon inhibition of TcREG9.1, which acts as a repressor of onward development. This release into the circulation, combined with the expression of genes associated with differentiation, would act to favour the parasites' transmission when ingested by tsetse flies sampling the circulatory blood. As in *T. brucei* some facets of the next developmental step to insect stages (kinetoplast repositioning) would also be regulated by TcREG9.1, although the overall impact of this would be dependent upon other signalling events received upon uptake into the fly. The identification

**Fig 9. Model for the consequences of TcREG9.1 depletion in *T. congolense* in vivo and upon development to insect forms.** *T. congolense* would replicate attached to the vasculature but upon TcREG9.1 RNAi, parasites detach and activate the expression of transcripts associated with development. The detached parasites can be taken up by the tsetse fly for transmission. In the fly, depletion of TcREG9.1 causes reduced kinetoplast repositioning ('KP'). However, in vitro this effect is overridden by the presence of the differentiation trigger cis aconitate, which activates both kinetoplast repositioning and PRS expression irrespective of the level of TcREG9.1.

of TcREG9.1 provides first insight into these adherence and developmental processes in *T. congolense*, unlocking understanding of the molecular events underlying these fundamental aspects of this parasite's biology.

## Materials and methods

### Ethical statement

Animal experiments in this work were carried out in accordance with the local ethical approval requirements of the University of Edinburgh (approval number PL02-12) and the UK Home Office Animal (Scientific Procedures Act (1986) under licence number PP2251183.

### Trypanosomes

*T. congolense* parasites of the IL3000 strain were used both for infections and in vitro experiments. This strain was derived from the ILC-49 strain that was isolated from a cow in the Trans Mara, Kenya [40]. *T. congolense* bloodstream forms were cultured in TcBSF3 supplemented with 20% goat serum and 1% Penicillin/ Streptomycin at 34°C with 5% CO2 [41]. *T. congolense* parasites were cultured up to a density of $1\times10^7$ cells/ml, and were generally not maintained at a density lower than $5\times10^4$ cells/ml, as parasites did not respond well to low density. Parasites were frozen at a density of $3\text{-}6\times10^6$ parasites in 1ml of freezing mix (TcBSF3 with 30% glycerol). Frozen stocks were stored at -80°C. When recovering parasites from frozen, thawed parasites were first pelleted and resuspended in 2ml fresh TcBSF3 to remove glycerol.

To assess the level of cell attachment and detachment, after gently swirling the flask back and forth, 10μl of supernatant was scored on a haemocytometer to provide a measure of detached cells. Attached cells were then dislodged by flushing 5-10ml of medium using a pipette gun, cell detachment being checked by microscopy. 10μl of the cell medium was then scored using a haemocytometer to give a total number of attached and detached cells, with the attached cells and detached cell values being determined from the cell count before and after flushing.

RNAi to TcREG9.1 was achieved by inserting a PCR fragment of the TbREG9.1 orthologue, TcIL3000.11.14540 (TCREG9.1 RNAi F 5'TCTAGAACTAGTACACATCGATG; TCREG9.1 ATCGATAAGCTTCAGGTGCGGCT) into p3T7-TcoV plasmid before transfection into the single marker line TcoSM, according to the methods described in [24]. The selection for transformants involved growth of parasites under G418 selection (0.5μg/ml) with puromycin (0.5μg/ml). For assessment of knockdown, two primer sets were used: 'new 2' (2F 5'CACAATGCCTAGTGGTAACG3'; 2R 5'CATCAAACACACACTCATTCC) and 'new 3' (3F 5'AGAGTGCTATGTGATTCCTCTT3'; 3R CGTTAACTCCTCCAAAACCA3').

### RNAseq analyses

RNA samples (combining attached and free swimming cells) were isolated from TcREG9.1 clones 3.3 and 4.5 induced with doxycycline or not for 3 days using QIAgen RNA preparation kit according to the manufacturer's instructions, including on-column DNAse treatment. Purified samples were assessed for integrity using a formaldehyde agarose gel and then shipped on dry ice to BGI Tech solutions in Hong Kong for RNA-seq analysis. RNA knockdown of the TcREG9.1 transcript was validated in each case using qRT-PCR as described above. The quality of the transcriptome data produced by BGI Hong Kong was evaluated using the Fast QC report programme (http://www.bioinformatics.babraham.ac.uk/projects/fastqc/). The reads were aligned to the *T. congolense* IL3000 genome using the Bowtie 2 programme (http://

bowtie-bio.sourceforge.net/bowtie2/index.shtml [42]. A group-wise comparison using DESEQ2 [43] was performed between induced and uninduced TcReg9.1 RNAi replicates.

## Cytological analysis

For the assessment of cell cycle status, parasites were airdried on to glass slides and fixed in ice cold methanol for at least 30 minutes. Thereafter, following rehydration in PBS, samples were incubated with 4',6-diamidino-2-phenylindole (DAPI) (10 μg ml$^{-1}$ in PBS) for 2 minutes and then washed for 5 min in PBS. Slides were then mounted with 40 μl Mowiol containing 2.5% 1,4-diazabicyclo(2.2.2)octane (DABCO). Cell cycle status was assessed by scoring the number of kinetoplasts and nuclei within each cell when visualised under a Zeiss Axio Imager Z2 mounted with a Prime BSI (Teledyne Photometrics) camera using a phase contrast objective (x40, x100).

## Cell motility analysis

Cells were grown to a density of $1x10^6$ cells/ml. 5 ml of these cells were then transferred to 6 well plates and incubated at 34°C, 5% $CO_2$ for 60 minutes to let the cells attach to the bottom of the wells. Thereafter, the cells were treated with DMSO, 10μM compound A and at various time points, cells were visualised at 40x magnification using a Olympus CKX53 microscope. Time-lapse images were generated with a QImaging Retiga-2000R digital camera. 100 images were then taken, with one image captured every 0.25 seconds and these images were imported into FIJI [44] and MTrackJ [45] was used to track the movement of 10 individual randomly chosen cells and the measurement of track statistics. A summary of each track was then exported for each cell line for further analysis in GraphPad Prism10.

## Differentiation assays

For differentiation analyses, parasites were incubated in DTM medium at 27°C in the presence or absence of 6mM cis aconitate. Reactivity of PRS antibody was determined using air dried methanol fixed samples reacted with anti-PRS antibody (a kind gift of Peter Butikofer, University of Bern Switzerland)(1: 200 dilution in PBS) followed after washing with PBS with anti-rabbit Alexa488 conjugated secondary antibody and then 4',6-diamidino-2-phenylindole (DAPI) (100 ng.mL-1) for 2 minutes. Slides were mounted in Fluorimount-G (Invitrogen). The kinetoplast-posterior dimension was determined using ImageJ on a Zeiss Axio Imager Z2 mounted with a Prime BSI (Teledyne Photometrics) camera using a phase contrast objective (x40, x100).

## Supporting information

**S1 Fig. *T. congolense* encode an orthologue of TbREG9.1.** The TcREG9.1 gene encodes a 519 amino acid protein with an RRM predicted RNA binding domain positioned centrally (yellow shading). The phylogenetic relationship between REG9.1 orthologues in different kinetoplastids is shown below the alignment.
(TIF)

**S2 Fig. Infected:control LC-MS/MS ratio of antigens recognized by IgG derived from animals at day 28 of infection by *T. congolense* strain 02J, 31J, KONT 2/133 and KONT2/151 based on Table S1 of Fleming et al., 2014 [28], and summarized in S1 Table.** Individual transcripts are colour coded according to their encoded protein family, with the gene code for the most differentially detected annotated.
(TIF)

**S3 Fig. Individual data of the development of parasites from each mouse (M1-M8) with or without REG9.1 depletion by RNAi, activated by inclusion of doxycline (M5-M8).** For each infection, harvested parasites were exposed or not to 6mM cis aconitate and the kinetoplast to posterior dimension determined at 48hr. As controls, cultured bloodstream forms (BSF) and cultured procyclic forms (PCF) were also included. The depletion of TcREG9.1 by RNAi resulted in a reduced kinetoplast repositioning in each case in the absence of cis aconitate.
(TIF)

**S1 Table. Read values and infected:control LC-MS/MS ratio of antigens recognized by IgG derived from animals at day 28 of infection by *T. congolense* strain 02J, 31J, KONT 2/133 and KONT2/151 based on Table S1 of Fleming et al., 2014 [28].** Individual transcripts are colour coded according to their encoded protein family.
(TIF)

**S2 Table. Pairwise statistical comparisons of data presented in Fig 8.**
(XLSX)

**S1 Data. RNAseq data for relative gene expression for bloodstream form *T. congolense* induced to knockdown TcREG9.1 or not.** Individual sheets provide values for distinct RNAi clones (clone 3.3; clone 4.5) and include expression values for all genes, or where the logFC change is >1 or <-1 (upregulated genes or downregulated genes). Sheets with transcripts upregulated in both clones, or down regulated in both clones are included, along with transcripts that show opposing direction of regulation in the respective clones. Transcripts that are upregulated or downregulated and have prediction for addition of a GPI anchor (potentially indicating surface expressed proteins) are also shown in separate sheets.
(XLSX)

## Acknowledgments

We thank Peter Bütikofer for the gift of antisera to PRS.

## Author Contributions

**Conceptualization:** Eleanor Silvester, Balazs Szoor, Keith R. Matthews.

**Data curation:** Eleanor Silvester, Balazs Szoor, Alasdair Ivens.

**Formal analysis:** Eleanor Silvester, Balazs Szoor, Alasdair Ivens, Keith R. Matthews.

**Funding acquisition:** Catarina Gadelha, Bill Wickstead, Keith R. Matthews.

**Investigation:** Eleanor Silvester, Balazs Szoor, Alasdair Ivens.

**Methodology:** Eleanor Silvester, Balazs Szoor, Georgina Awuah-Mensah, Catarina Gadelha, Bill Wickstead.

**Project administration:** Catarina Gadelha, Bill Wickstead, Keith R. Matthews.

**Software:** Alasdair Ivens.

**Supervision:** Catarina Gadelha, Bill Wickstead, Keith R. Matthews.

**Validation:** Eleanor Silvester, Balazs Szoor, Alasdair Ivens, Keith R. Matthews.

**Writing – original draft:** Eleanor Silvester, Balazs Szoor, Alasdair Ivens, Catarina Gadelha, Bill Wickstead, Keith R. Matthews.

**Writing – review & editing:** Eleanor Silvester, Balazs Szoor, Alasdair Ivens, Catarina Gadelha, Bill Wickstead, Keith R. Matthews.

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
