## [Decision Letter · Decision Letter 0]

29 Jan 2024

Dear Prof. Matthews,

Thank you very much for submitting your manuscript "A conserved trypanosomatid differentiation regulator controls substrate attachment and morphological development in Trypanosoma congolense" for consideration at PLOS Pathogens. As with all papers reviewed by the journal, your manuscript was reviewed by members of the editorial board and by several independent reviewers. The reviewers appreciated the attention to an important topic. Based on the reviews, we are likely to accept this manuscript for publication, providing that you modify the manuscript according to the review recommendations.

The manuscript was considered important and interesting by all reviewers and it requires only minor revisions/clarifications. Please take into consideration these comments and resubmit a new version.

Sincerely,

Roberto Docampo

Academic Editor

PLOS Pathogens

Jeffrey Dvorin

Section Editor

PLOS Pathogens

Michael Malim

Editor-in-Chief

PLOS Pathogens

orcid.org/0000-0002-7699-2064

The manuscript was considered important and interesting by all reviewers and it requires only minor revisions/clarifications. Please take into consideration these comments and resubmit a new version.

Reviewer Comments (if any, and for reference):

Reviewer's Responses to Questions

**Part I - Summary**

Reviewer #1: This is an interesting and important study by Silvester and Szoor et al, demonstrating that TcREG9.1, an orthologue of the T. brucei TbREG9.1 which regulates ESAG9 expression and some aspects of differentiation in T. brucei, plays a similar role in T. congolense, despite the absence of a morphologically distinct transmission-adapted stage in this parasite species. Moreover, they show that detachment from the microvasculature plays an important role in facilitating transmission for this parasite. Overall, this work represents an important advance in understanding how T. congolense adapts for transmission and the conserved nature of this process in African trypanosomes. The experiments are straightforward and well-controlled. I only have minor comments, primarily relating to the clarity of the discussion around the RNA-seq data. These are outlined below.

Reviewer #2: SYNOPSIS: Authors identify a T congolense homologue of REG9.1, a regulator of development in T brucei. They find that KD (~50% mRNA levels) of TcREG9.1 reduced growth rate, caused increased motility and reduced attachment to substrate surface in vitro. Noteably, the attachment and motility changes were phenocopied by PDE inhibition, indicating involvement of cAMP, as reported for attachment of the related parasite, Crithidia. RNAseq analysis revealed that TcREG9.1 depletion leads to misregulation of genes associated with T congolense development and parasite density-dependent responses in the mammalian bloodstream. Analysis of mouse-derived parasites corroborated this finding by revealing that some changes in cell morphological markers are TcREG9.1-dependent. Finally, the number of circulating parasites in an animal infection model was also increased in the TcREG9.1 KD, interpreted as consistent with anticipated reduced attachment to blood vessels.

CRITIQUE: This work addresses an important pathogen, T. congelense, that is much less studied than the related T. brucei and therefore provides considerable novelty and significance. The work is done rigorously, with suitable replicates, including independent isolates of the KD line, and the conclusions are well-supported. The paper is written clearly and concisely. I only have a few minor questions/comments for the authors to consider:

QUESTIONS/COMMENTS:

• Growth curves indicate growth subsides, then returns to approximately control growth rate. Does this reflect RNA returning to control levels, or parasite adapting to reduction of mRNA in the KD?

• Fig 2. I didn't see description of how % parasites in supernatant was determined. I imagine one can count parasites in the supernatant, but what method is used to count attached parasites to get the total number?

• Regarding impact of PDE-inhibition on attachment - can the authors distinguish an effect on attachment, per se, versus an indirect effect, namely reduced attachment as a result of more vigorous motility? If not, text should be adjusted to indicate this. Note also that there is work in T brucei indicating loss of PDE results in reduced turning and greater straight-line velocity [Shaw et al. 2019 Nat Commun 10, p803], perhaps analogous to the increased motility seen for T. congolense here.

• For RNAseq analysis:

○ -- were these analyses done on detached cells only, or the entire population?

○ -- I see logFC threshold. It would be helpful to include the adj p-value threshold (per leged) on the figure itself.

• Regarding anticipated surface location based on surface phylome annotation, the authors might also consider cross-referencing with the stage-dependent surface proteome analysis done in T. brucei [Shimogawa et al. (2015)Mol Cell Proteomics. 14, p1977], which looks at surface proteins directly, rather than predicting surface location via sequence analysis.

• Fig 7: The figure legend is unclear.

○ Does each plot correspond to a different animal?

○ It would add clarity to indicate "Uninduced" and "Induced" directly on panel a and b, respectively.

○ Update the 7b legend to change ndicates a and b are (a) vs "TcREG9.1 cell line" to "TcREG9.1 RNAi cell line".

• Regarding Fig 8.

○ Panel C: I would recommend kp-posterior dimension cartoon trypanosomes be black, since the data in that chart do not reflect PRS expression, which I interpret to be represented by red vs blue outline. Likewise, the PRS panel should exclude the nucleus and kp position, as those are not reflected in the data shown there. It would also be helpful to explain this in the figure legend, which currently does not mention the cartoon trypanosomes.

○ It would be useful (though not required) if the authors did this experiment looking at parasite attachment directly, as done in the orginal Banks paper. I agree, however, that the experiment as done would appear to provide an indirect proxy for estimating attachment. Looking back at the abstract, they phrase the interpretation appropriately.

§ NOTE: Has a similar analysis been done with T. brucei? If the difference between TS and CP reflects attachment to vessels exclusively, the ratio should approximate unity for T. brucei, correct?

○ For the claim on p13 that "…mouse derived parasites responded to cis aconitate after 24h by increasing their kinetoplast posterior dimension", please provide statistical analysis of the 24hr data on kp-posterior distance.

Reviewer #3: The experiments in the manuscript build on those using Trypanosoma brucei that have shown that there is a clear developmental transition in bloodstream form trypanosomes from proliferative to a morphologically distinct cell cycle arrested stumpy form. These experiments identified mRNAs that are increased in stumpy forms including REG9.1 which encodes a protein containing a RRM motif, often but not always involved in RNA binding.

Here, the T. congolense homologue of REG9.1 is analysed and RNAi-mediated depletion shown to alter the ability to sequester onto surfaces. Cell lines were constructed to knock down REG9.1 mRNA and resulted in a ~50% decrease. The most noticeable phenotype was a reduction in adherence. This was phenocopied by a phosphodiesterase inhibitor which also altered the motility (distance travelled). The experiments go on to characterise changes in mRNA levels after 3 days of RNAi and provide strong evidence it represents a similar pathway to that present in T. brucei.

Overall, the experiments show that a similar developmental pathway regulates stumpy formation in T. brucei and adherence in T. congolense. It is a good piece of work.

**Part II – Major Issues: Key Experiments Required for Acceptance**

Reviewer #1: (No Response)

Reviewer #2: NA

Reviewer #3: none

**Part III – Minor Issues: Editorial and Data Presentation Modifications**

Reviewer #1: Figure 6 - The discussion around Figure 6 was hard to follow for me. Family 22 is highlighted but expression is really variable; how is this supposed to be interpreted? I was also confused by the focus on immunodominant antigens?. There could be more discussion of what this might mean. Why would transmission associated antigens be immunodominant? Moreover, the variability among members of families that are highlighted is confusing. As it is, I am not sure what this analysis adds to the manuscript. Therefore, I'd recommend more discussion/context to make the implications of the highlighted findings absolutely clear. Looking at the discussion, it appears the major observation is that surface proteins generally are upregulated after knockdown. Perhaps some statistical analysis would show they are disproportionately represented in the upregulated gene set?

Figure 8 - While everything is in this figure, I found it hard to link the data in the figure to the descriptions in the text. There may be ways to reorganize the visualization so that the pertinent comparisons are highlighted and more obvious to a reader (for example, putting samples/conditions you explicitly contrast next to one another on the graph). This is not a major issue, but it did take me a while to parse this particular figure.

Figure 9 - I assume KP stands for kinteoplast position. The authors could spell this out on the figure or in the legend.

Lines 84-85 - It may be more clear to state that the two morphological forms mentioned are in the mammalian host. This is mentioned in later sentences but I think better to state up front.

Line 113 - RRM is defined eventually on line 389 but should probably be defined at the first mention

Line 174 - typo, I think you mean "knock down"

Reviewer #2: See Summary, "Questions/Comments"

Reviewer #3: Define ‘immunodominant’ in abstract. Experimental rather than annotation and animal in which they are immunodominant.

Line 180

Comment on whether the reduction in growth is transient for ~3 days

Line 212

Comment on high concentration of PDE inhibitor, was it the same as used for the experiments in Crithidia?

PLOS authors have the option to publish the peer review history of their article (what does this mean?). If published, this will include your full peer review and any attached files.

Reviewer #1: No

Reviewer #2: **Yes: **Kent L. Hill

Reviewer #3: No

Figure Files:

Data Requirements:

Reproducibility:

References:

---

## [Editor Report · Decision Letter 1]

8 Feb 2024

Dear Prof. Matthews,

We are pleased to inform you that your manuscript 'A conserved trypanosomatid differentiation regulator controls substrate attachment and morphological development in Trypanosoma congolense' has been provisionally accepted for publication in PLOS Pathogens.

Best regards,

Roberto Docampo

Academic Editor

PLOS Pathogens

Jeffrey Dvorin

Section Editor

PLOS Pathogens

Michael Malim

Editor-in-Chief

PLOS Pathogens

orcid.org/0000-0002-7699-2064

Authors have responded satisfactorily to the reviewers' questions.
---

## [Editor Report · Acceptance letter]

22 Feb 2024

Dear Prof. Matthews,

We are delighted to inform you that your manuscript, "A conserved trypanosomatid differentiation regulator controls substrate attachment and morphological development in *Trypanosoma congolense*," has been formally accepted for publication in PLOS Pathogens.

Best regards,

Michael Malim

Editor-in-Chief

PLOS Pathogens

orcid.org/0000-0002-7699-2064